# Two Rare Variants in *PLAU* and *BACE1* Genes—Do They Contribute to Semantic Dementia Clinical Phenotype?

**DOI:** 10.3390/genes12111806

**Published:** 2021-11-17

**Authors:** Katarzyna Gaweda-Walerych, Emilia J. Sitek, Małgorzata Borczyk, Mariusz Berdyński, Ewa Narożańska, Bogna Brockhuis, Michał Korostyński, Jarosław Sławek, Cezary Zekanowski

**Affiliations:** 1Laboratory of Neurogenetics, Mossakowski Medical Research Institute, Polish Academy of Sciences, 02-106 Warsaw, Poland; m.berdynski@imdik.pan.pl (M.B.); c.zekanowski@imdik.pan.pl (C.Z.); 2Neurology Department, St. Adalbert Hospital, Copernicus PL, 80-462 Gdansk, Poland; gdansk.ewa1@gmail.com (E.N.); jaroslaw.slawek@gumed.edu.pl (J.S.); 3Division of Neurological and Psychiatric Nursing, Faculty of Health Sciences, Medical University of Gdansk, 80-462 Gdansk, Poland; 4Laboratory of Pharmacogenomics, Department of Molecular Pharmacology, Maj Institute of Pharmacology Polish Academy of Sciences, 31-343 Krakow, Poland; gosborcz@if-pan.krakow.pl (M.B.); michkor@if-pan.krakow.pl (M.K.); 5Division of Nuclear Medicine, Faculty of Health Sciences, Medical University of Gdansk, 80-214 Gdansk, Poland; bognabro@gumed.edu.pl

**Keywords:** whole-genome sequencing (WGS), urokinase-type plasminogen activator (PLAU) haploinsuficiency, β-site APP-cleaving enzyme 1 (BACE1), mtDNA polymerase gamma (POLG), semantic dementia, atypical Alzheimer’s disease, primary skin fibroblasts, magnetic resonance imaging (MRI), single-photon emission computed tomography (SPECT)

## Abstract

We have performed whole-genome sequencing to identify the genetic variants potentially contributing to the early-onset semantic dementia phenotype in a patient with family history of dementia and episodic memory deficit accompanied with profound semantic loss. Only very rare variants of unknown significance (VUS) have been identified: a nonsense variant c.366C>A/p.Cys122* in plasminogen activator, urokinase (PLAU) and a missense variant c.944C>T/p.Thr315Met in β-site APP-cleaving enzyme 1 (BACE1)—along with known disease-modifying variants of moderate penetrance. Patient-derived fibroblasts showed reduced *PLAU* and elevated *BACE1* mRNA and protein levels compared to control fibroblasts. Successful rescue of *PLAU* mRNA levels by nonsense-mediated mRNA decay (NMD) inhibitor (puromycin) confirmed NMD as the underlying mechanism. This is the first report of the *PLAU* variant with the confirmed haploinsufficiency, associated with semantic dementia phenotype. Our results suggest that rare variants in the *PLAU* and *BACE1* genes should be considered in future studies on early-onset dementias.

## 1. Introduction

Clinical diagnosis of dementia is often difficult and inconclusive due to overlapping clinical presentations and neuropathology [1,2]. Moreover, the implementation of next-generation sequencing methods revealed a complex genomic nature of disease phenotypes, challenged the classical definition of genetic causality, and the concept of strictly monogenic disorders, shifting the pathogenicity model from monogenic to oligogenic inheritance [3,4].

Here we present a patient with the phenotype of early-onset semantic dementia. The diagnosis of atypical Alzheimer’s disease (AD) was also considered due to episodic memory deficit accompanying profound semantic loss. Whole-genome sequencing (WGS) analysis did not reveal any pathogenic/likely pathogenic variants, as defined by The American College of Medical Genetics and Genomics (ACMG) criteria. Thus, we have selected the most promising variants of unknown significance (VUS) in *PLAU* (Plasminogen Activator, Urokinase) and *BACE1* (β-site APP-cleaving enzyme 1) genes for which we performed functional in vitro analysis and discussed their putative contribution to the observed clinical phenotype.

## 2. Materials and Methods

### 2.1. Whole-Genome Sequencing

Genomic DNA was extracted from peripheral blood leukocytes using a standard salting-out procedure [5]. Then WGS was performed (Novogene, Beijing, China) according to the following protocol: sequencing libraries were generated using NEBNext Ultra II DNA Library Prep Kit for Illumina (New England Biolabs, Hitchin, UK) following the manufacturers’ recommendations. Genomic DNA was randomly fragmented to a size of 350 bp by Bioruptor, DNA fragments were size-selected with sample purification beads. The selected fragments were end-polished, A-tailed, and ligated with the full-length adapter. After the treatments, the fragments were filtered with beads again. Finally, the libraries were analyzed for size distribution by Agilent2100 Bioanalyzer and quantified using real-time PCR. Libraries were sequenced by Illumina high-throughput HiSeq X Ten sequencer with paired-end sequencing strategy.

### 2.2. WGS Data Preprocessing

Raw read files were processed with Intelliseq Flow Annotation Pipeline (https://intelliseq.com/, accessed on 31 July 2021) based on Cromwell (https://cromwell.readthedocs.io/en/stable/, accessed on 31 July 2021). Within the pipeline, fastq file quality was assessed with FastQC. Further, fastq files were then aligned to Broad Institute Hg38 Human Reference Genome with GATK 4.0.3. Duplicate reads were removed with Picard and base quality Phred scores were recalibrated using GATK’s covariance recalibration. Variants were called using GATK best practices. Identified variants were assessed using the Intelliseq Flow annotation pipeline that implemented ACMG recommendations.

### 2.3. Quality Assessment

Mean sequencing depth for the whole genome was 24.95 and the mean genotype quality (GQ) score for the whole-genome vcf was 83.75. 99.9% of genotypes had an estimated probability of error <10%. Additional quality analysis was conducted for each of the genes included in the panel (for both selected rare and all 368,460 variants identified in the panel genes) and is available as Appendix A.

### 2.4. Rare Variant Analysis

Whole-genome sequencing identified 4,827,282 SNVs and small indels in the sample. Further filtering was based on Phred quality scores, allele frequency in the ExAC (Exome Aggregation Consortium) database (<5% for variants in genes associated with the disease phenotype), association with Human Phenotype Ontology (HPO) terms, and predicted pathogenicity. Variants with the following impact on protein and mRNA were retained: missense, nonsense, frameshift, and splice site variants. Common and low impact variants were then filtered out (with a max frequency threshold of 0.05 and minimal SnpEff predicted impact on gene product set as MODERATE). Low-quality multiallelic variants (QUAL < 300) were removed.

Annotated variants for genes from the defined gene list (Appendix A) were then classified according to the American College of Medical Genetics and Genomics (ACMG) criteria and prioritized. Selected variants were manually evaluated for quality in IGV. *PLAU* and *BACE1* variants were confirmed with Sanger sequencing (ABI 3130 Genetic Analyzer, Applied Biosystems, Foster City, CA, USA) (Appendix A).

### 2.5. Copy Number Variation Analysis

Structural variants (SVs) in selected gene panel were called with Intelliseq Flow Structural Variants pipeline (https://intelliseq.com/, accessed on 14 August 2021). Within the pipepline variants were called with lumpy/smoove (https://github.com/brentp/smoove, accessed on 14 August 2021) [6] and annotated with duphold [7]. SVs were then filtered with the following parameters: QUAL > 30, del_cov_max = 0.7, dup_cov_min = 1.3, min_snp_count = 4, het_max = 0.25.

### 2.6. Mutation Screening of the PGRN, MAPT and C9orf72 Genes

Sanger sequencing, genotyping and expansion analysis were performed using previously described protocols [8,9,10] using Genetic Analyzer 3130 and SeqScape v2.5 software (Applied Biosystems).

### 2.7. Fibroblast Cultures and Inhibition of Nonsense-Mediated Decay (NMD)

Primary skin fibroblasts (obtained from the patient and age-matched, unrelated neurologically healthy subjects—Appendix A) were collected and cultured as previously described [11]. For NMD analysis primary fibroblast cultures were treated with an NMD inhibitor, puromycin (100 µg/mL or water) (Sigma Aldrich, Saint Louis, MO, USA) for 8 h [12,13].

### 2.8. mRNA Expression Analysis—Real Time PCR

RNA was extracted and reverse transcribed according to standard protocol with QIAzol Lysis Reagent (Qiagen, Manchester, UK) and QuantiTect^®^ Reverse Transcription kit (Qiagen, Manchester, UK), respectively. Quantitative real-time PCR analysis was done with RT HS-PCR Mix SYBR (A&A BIOTECHNOLOGY, Gdańsk, Poland) (primers are listed in Appendix A), using a StepOne Plus system (Applied Biosystems, Foster City, CA, USA). Changes in gene expression were determined with the ∆Ct method using *GAPDH* levels for normalization. Similar results were obtained with *PPIB* as a normalizing gene.

### 2.9. Western Blot

Western blot was performed as previously described [11] with the following primary antibodies: PLAU (Abcam, catalog number: ab169754, 1:1000), BACE1 (Cell Signaling, catalog number: 5606, 1:500), GAPDH (Merck Millipore, catalog number: MAB374, 1:20,000). The representative experiment result is shown.

### 2.10. Statistical Analysis

For each experiment, the relative values obtained from different biological replicates (*n* ≥ 3) were used to calculate mean, standard deviations (SD) and statistical significance in a two-tailed *t*-test (GraphPad Prism 6.0). *p* < 0.05 was considered significant. For quantitative densitometric analysis of WB results, ImageJ software was used [14]. The intensity value of each protein band was normalized to the respective GAPDH value.

Extended methods can be found in Appendix A.

## 3. Results

### 3.1. Case Description

A 69-year-old patient, with university education and professional experience in forestry and agriculture, was referred to our neurology outpatient clinic with a five year history of cognitive dysfunction. When he was 64 (age at onset), he started experiencing problems with planning chemical fertilizers’ application while running a large agricultural company. His insight was initially preserved as he employed a new worker to help him with the tasks he could no longer perform on his own. Progressive cognitive decline was evident in the report provided by his family. He was no longer able to recognize fruit types, he could not name trees or birds, despite his previous vast knowledge in this area. He became very effusive, also to strangers. The patient had a strong family history of dementia (Figure 1A). One of the proband’s paternal aunts was diagnosed with an unspecified late-onset dementia that also progressed slowly. One of the proband’s brother presented with a mild cognitive impairment with predominant semantic deficit. The results of his neuropsychological examination are presented in a Appendix A. Both the proband’s father and the above-mentioned brother had marked cognitive rigidity (Figure 1A).

At the time of the first clinical evaluation the general neurological examination did not reveal any motor or sensory problems nor frontal release signs. There was no evidence of depression or psychotic symptoms. His speech was fluent and effortless but empty. He did not demonstrate any insight.

The neuropsychological assessment planned and executed by a neuropsychologist showed dementia (Addenbrooke’s Cognitive Examination III: 59/100) (Appendix A), which was consistent with the patient’s inability to perform complex daily life activities. His speech was severely anomic. Phonology, syntax and prosody were not affected. Phonemic fluency was initially preserved in contrast to very deficient semantic fluency (Appendix A). He had a widespread semantic deficit, which affected both verbal and visual semantics. He was unable to recognize many very famous faces. His spatial skills were very well preserved which contrasted with a very poor semantic memory as demonstrated by impaired drawing to command but good copying (Figure 1C). He had episodic memory impairment and executive dysfunction, while the calculation was very well preserved [15]. Praxis was unaffected in copying tasks that did not require reference to semantics.

The pattern of progression in neuropsychological assessment, as well as behavioral deterioration over the next years was typical for semantic dementia/semantic variant of a primary progressive aphasia [16,17]. He remained a good swimmer and dancer during the next few years. He continued to enjoy numerical paper and pencil games, especially Sudoku. However, he could no longer handle computer games. At the age of 71, he could not understand Easter celebrations nor understand the word “glass”. Face recognition deficits progressed. At the age of 72 semantic deficit progressed significantly e.g., he was unable to recognize snow and tried to eat a dishwasher tablet. Instead of using his wife’s first name, he called her “my girl”. Parsimony appeared. He also required constant supervision due to compulsive behavior and disinhibition, e.g., during a walk he stole chips from a child who was passing by. He was also very agitated and had transient psychotic symptoms. At this time neuropsychological assessment had to be shortened due to impatience and deficient verbal comprehension. As he required 24h supervision and nursing care due to severe dementia and incontinence, he was institutionalized at the age of 73 and deceased in September 2021 at the age of 76.

MRI performed at the age of 67 and 69 revealed mild bilateral temporal atrophy that became more pronounced on the right side at the age of 72 (Figure 1B, left panel)

Single-photon emission computed tomography (SPECT) was conducted when he was 70 and showed an anterior pattern of hypoperfusion with bilateral frontal and temporal involvement with more pronounced deficits in the superior frontal gyrus and the temporal lobe on the right side (Figure 1B, right panel).

### 3.2. Genetic Analyses

Initial genetic analysis of the patient’s DNA has excluded mutations in *MAPT*, *PGRN*, and *C9orf72* which are the most common genetic causes of the FTLD spectrum [18] and subsequently whole-genome sequencing was performed. Neither pathogenic or likely pathogenic variants (P/LP variants) nor high-confidence structural variants have been identified according to the ACMG criteria for pathogenicity [19]. However, very rare variants of unknown significance (VUS) have been found in *PLAU*, *BACE1*, and *POLG* genes along with common established risk factors in *APOE* and *MAPT* that could contribute to the observed phenotype (Table 1, and Appendix A).

#### 3.2.1. PLAU (Urokinase-Type Plasminogen Activator)

PLAU is a secreted serine protease that converts plasminogen to plasmin, triggering the downstream fibrinolysis cascade (#191840, OMIM). The mutation in the *PLAU* gene (NM_002658.6 ENST00000372764.4, c.366C>A;p.Cys122*) (Table 1, Appendix A) has been classified as a high confidence loss of function variant with the ACMG score of 0.6 (Appendix A). The *PLAU* c.366A has been previously reported only in the ALFA (for Alzheimer and Families) project (MAF = 0.00005; 1/21336), a research platform to identify early pathophysiological features of AD [20]. *PLAU* c.366C>A was absent from gnomAD (11.10.2021) and in Polish WGS samples (*n* = 287, an in-house database): sportsmen (*n* = 102); patients with Tourette syndrome (*n* = 129), and healthy controls (*n* = 56).

*PLAU* position c.366 is highly evolutionarily conserved according to PhastCons (1.0; high: >0.93) and GERP = 5.68 (high: >3.26) and C>A substitution introduces a stop codon in exon 5 (out of 11). As a consequence, a fraction of transcripts is predicted to undergo nonsense-mediated mRNA decay (NMD) which was confirmed by the successful rescue of *PLAU* mRNA levels (Figure 2A) and PLAU mutant allele (Appendix A) by an NMD inhibitor, puromycin [12,13]. Accordingly, the *PLAU* mRNA and protein levels were decreased in the patient’s fibroblasts compared to neurologically healthy, age-matched control individuals (Figure 2A,B).

#### 3.2.2. BACE1 (β-Site APP-Cleaving Enzyme 1, Beta-Secretase 1)

BACE1 (# 604252, OMIM) is a transmembrane aspartic protease that catalyzes the first step in the formation of amyloid-beta (Ab) peptides from amyloid precursor protein (APP) [21]. Amyloid-beta peptides build amyloid-beta plaques that accumulate in the brains of AD patients [21]. Since BACE1 levels/activity are elevated in AD brains and fluids, the inhibition of this β-secretase has been extensively tested in clinical trials as a therapeutic strategy for AD [21]. c.944C>T variant (NM_012104.6 ENST00000313005.11, Table 1, Appendix A) was predicted to be pathogenic by six programs: DANN, EIGEN, FATHMM-MKL, LIST-S2, M-CAP, and MutationTaster, and to affect splicing (Human Splicing Finder) (Appendix A). There was no evidence of additional/aberrant mRNA isoforms (Appendix A). However, we have found elevated *BACE1* mRNA and protein levels in patient’s fibroblasts compared to controls (Figure 2C,D).

Other identified variants potentially contributing to the clinical phenotype are presented in Appendix A, filtering criteria are described in Materials and Methods.

## 4. Discussion

Overlapping features of dementia phenotypes pose a considerable challenge for clinicians, especially semantic dementia, which in most cases is considered to be sporadic [22].

In our patient prominent deficits in both verbal and visual semantics, face recognition and episodic memory were present, consistent with the clinical presentation of a right temporal variant of frontotemporal dementia (rtvFTD) [23]. However, the changes observed in neuroimaging were bilateral with only slight right-sided predominance, so his clinical-radiological presentation did not fully correspond to the rtvFTD. The pattern of semantic impairment in this patient showed that limiting SD to a semantic variant of a primary progressive aphasia (svPPA) according to Gorno-Tempini et al. criteria [24] obscures the global semantic deficit. Although some patients may be classified as svPPA or rtvFTD our case presented with a mixed pattern of deficits, corresponding to the previous Lund-Manchester criteria of SD [25].

The disease duration was relatively long in the proband (12 years) and his affected relatives. Of note, long disease duration is typical for SD [26], unlike other early-onset dementias, which have a rather fast slope of progression.

The possible hereditary nature of semantic dementia has been recently shown in rtvFTD [27]. The potential association of psychotic symptoms in one of the proband’s brothers with FTD cannot be excluded, although in FTD psychotic symptoms are usually associated with *C9orf72* mutation [28].

While the bioinformatic analysis of WGS results revealed no pathogenic or likely pathogenic variants, in vitro functional analyses suggested that the very rare variants of unknown significance in *PLAU* (c.366C>A; p.Cys122*), and *BACE1* (c.944C>T; p.Thr315Met) could contribute to the disease phenotype.

To date, only heterozygous tandem duplications of *PLAU* have been found to cause the disease phenotype of autosomal dominant Quebec platelet disorder (QPD, # 601,709 OMIM). QPD is a bleeding disorder due to a gain-of-function defect in fibrinolysis with significantly increased PLAU levels in patient’s platelets, who showed delayed onset bleeding after vascular damage (e.g., surgery). To our knowledge, p.Cys122* (Table 1), is the first PLAU variant with the confirmed haploinsufficiency detected in a patient presenting semantic dementia phenotype. The loss-of-function (LOF) variants of the *PLAU* gene have not been previously reported in humans, and the consequences of PLAU deficiency have been studied only in mouse models. Mice lacking *PLAU* showed a reduced spontaneous exploration of the surrounding environment, impaired post-injury recovery [29,30], and had elevated amyloid-beta (Ab42 and Ab40) levels in plasma [31]. *PLAU* deficiency in mice also led to delayed wound healing with an abnormal angiogenic pattern [32]. On the other hand, Ab aggregates induced *PLAU* expression leading to increased plasmin, which in turn, degraded both aggregated and non-aggregated forms of Ab suggesting a negative feedback loop [33].

Interestingly, genotypes C/T and T/T of a missense *PLAU* variant in exon 6 (c.422T>C, Leu141Pro, rs2227564) have been previously associated with elevated plasma Ab42 levels and the increased LOAD risk (late-onset AD) [31,34], although the data remained controversial [35,36] (see also meta-analysis on http://www.alzgene.org, accessed on 5 October 2021). While the Pro141Leu variant seems to bind fibrin aggregates more efficiently [37] the exact molecular mechanism underlying its action is not known. However, our patient carried a genotype C/C at the position c.422. Moreover, other rare missense and nonsense single nucleotide variants (SNVs) in *PLAU* (p.T86A, p.H149Y, and p.C151F) have been identified in patients with multiple sclerosis (MS), an inflammatory disease characterized by myelin loss and neuronal dysfunction [38]. Finally, PLAU has also been identified as a frailty biomarker in aging and age-related diseases [39]. It can be speculated that the decreased PLAU levels due to p.Cys122* variant could lead to the dysfunction of fibrinolysis cascade and inefficient blood clots dissolution. However, our patient did not have any history of thrombotic disorders.

While patients with SD showed predominantly TDP-43 type C brain pathology, other pathology types, i.e., FTLD-TDP types A and B, Tau-positive (FTLD-tau) and Alzheimer’s disease (AD) pathology have been reported in 17% up to 32% of SD cases [22,40]. A recent neuropathological report suggests that rtvFTD in particular may be caused by other pathologies apart from FTLD-TDP type C [41]. It highlights the distinction between svPPA and rtvFTD. Unfortunately, in our case the diagnosis of SD (rtvFTD) was not supported by a definite exclusion of Alzheimer’s disease pathology through cerebrospinal fluid biomarker testing, amyloid-PET (Positron Emission Tomography) or neuropathological examination. For this reason, it cannot be excluded that the patient had amyloid-beta neuropathology. That speculation is strengthened by the fact that he showed several features of atypical AD. It is worth noting that the increased BACE1 mRNA/protein levels (Table 1, Figure 2C,D) could possibly increase the amyloid-beta load and in this respect act in synergy with PLAU deficiency [31]. It is also worth mentioning that the patient’s other brother was diagnosed with paranoid schizophrenia (Figure 1A), in which BACE1 could play an important role [42]. To date, only the impact of common BACE1 polymorphisms on AD risk has been studied, yielding discordant results (see also meta-analysis on http://www.alzgene.org, accessed on 5 October 2021).

In addition, we report a *POLG* variant (mtDNA polymerase gamma; NM_002693.3 ENST00000268124.9, c.3436C>T, p.Arg1146Cys; Table 1), as it has been described previously in autosomal-dominant progressive external ophthalmoplegia type 1 (adPEO1) [43]. The pathogenic status of POLG p.Arg1146Cys variant is controversial and our patient has not shown any symptoms of adPEO1. However, since mitochondrial impairment is a common mechanism of many neurodegenerative disorders, it could be speculated that variants interfering with the mitochondrial function could aggravate the phenotype. This is also suggested by the fact that another *POLG* mutation (p.Y955C) has been associated previously with Alzheimer’s pathology [44].

Finally, known moderate-impact variants such as *APOE* (ε2, ε4), which translates into a threefold greater risk of developing AD by age of 75 y. [45], as well as *MAPT* H1/H1 haplotype, previously associated with an increased AD risk [46] (Table 1), may contribute to the observed clinical phenotype.

Although our functional analyses suggested that *PLAU* and *BACE1* variants had a biological impact, further functional and segregation studies are needed to clarify their pathogenic status. Overall, it could be envisioned that more complex patterns of inheritance including oligogenic inheritance may account for part of early-onset dementia cases. However, the discrimination between causative and modifying variants, and insignificant ones is one of the greatest challenges of medical genetics. The concept of oligogenic inheritance should influence not only the approach to gene identification but also genetic testing and counseling [47].

## 5. Conclusions

Our findings suggest that future genetic analyses of early-onset dementia cases, especially those with slow disease progression, should take into account very rare variants in the *PLAU* and *BACE1* genes, that both are responsible for the regulation of amyloid-beta levels. 

## Figures and Tables

**Figure 1 genes-12-01806-f001:**
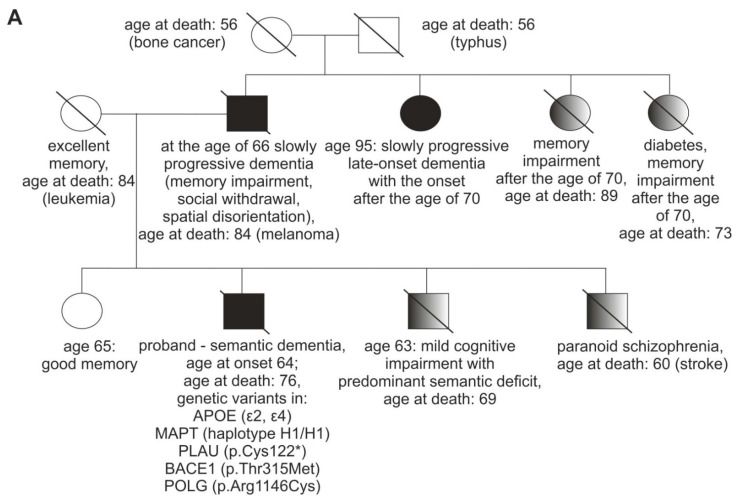
Family history, brain imaging of the patient, and drawing/calculation tasks. (**A**). Patient’s family history, shaded figures represent three family members who had some neuropsychiatric symptoms, but were not diagnosed with dementia; (**B**). Mild atrophy of right temporal lobe (arrow) on transaxial MRI image (left panel); Reduced perfusion in right temporal lobe (arrowhead) and slightly reduced perfusion, diffused in both frontal lobes in SPECT (right panel); Abbreviations: R- right, L-left; MRI, magnetic resonance imaging; (**C**). Drawing, copying, and calculation tasks demonstrating the dissociation between impaired semantics and well preserved visuospatial processing (inability to retrieve specific features of an object with good visuoperceptual and visuoconstructive functions) and calculation. I A. Drawing a bicycle to a verbal command at the age of 71. I B. Copy of a bicycle drawing at the age of 71. II. The calculation task was performed correctly at the age of 71, as well as at the age of 72. III A. Drawing a bicycle to a verbal command at the age of 72. III B. Copy of a bicycle drawing at the age of 72.

**Figure 2 genes-12-01806-f002:**
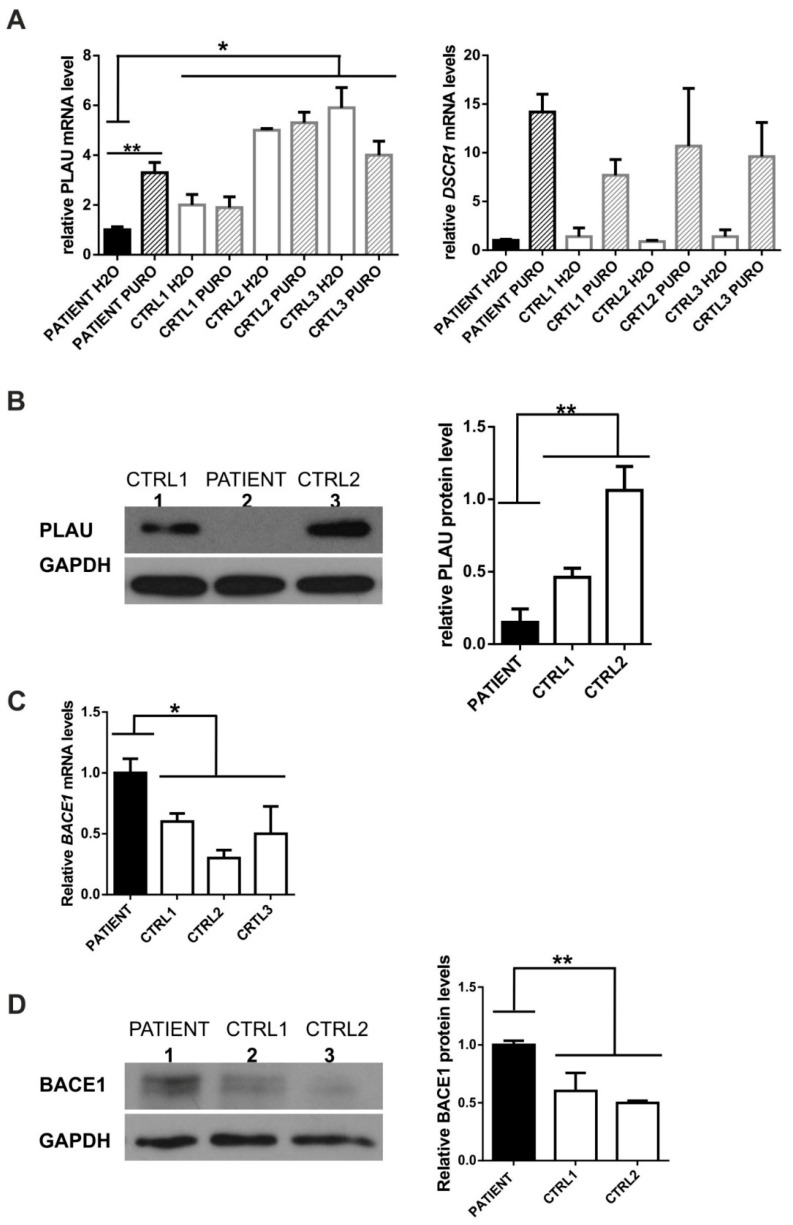
Functional analysis of PLAU and BACE1 variants in patients’ fibroblasts. (**A**) Decreased *PLAU* mRNA level in patient-derived fibroblasts (PATIENT) compared to control fibroblasts (CTRL1, CTRL2, CTRL3) could be rescued upon puromycin treatment (PURO) (left panel); As a positive control for NMD, the Down syndrome critical region 1 (*DSCR1*) gene was used (right panel); as a control for even cDNA input in RT-PCR *GAPDH* gene was used. (**B**) Decreased PLAU protein level in patient-derived fibroblasts (PATIENT) compared to control fibroblasts (CTRL1, CTRL2); densitometric analysis of PLAU protein level (*n* = 3), (right panel) (**C**) Increased *BACE1* mRNA level in the patient (PATIENT) compared to control fibroblasts (CTRL1, CTRL2, CTRL3); (**D**) Increased BACE1 protein level in the patient compared to control fibroblasts (CTRL1, CTRL2) (left panel), densitometric analysis of BACE1 protein level (*n* = 3), (right panel); * *p* < 0.05; ** *p* < 0.01.

**Table 1 genes-12-01806-t001:** The highest impact variants detected in the patient.

	Gene	HGVS ^1^ DNA/Protein	Predicted Effect	MAF gnomAD	CADD ^2^
Rare variants	*PLAU*	heterozygous c.366C>A/p.Cys122stop	mRNA nonsense mediated decay, haploinsuficiency	-	38
	*BACE1*	heterozygous c.944C>T/p.Thr315Met	missense (splicing variant)	0.00002389	29.5
	*POLG*	heterozygous c.3436C>T/p.Arg1146Cys	missense	0.00018695	35
Common variants	*APOE*	ε2/ε4	-	-	-
	*MAPT*	H1/H1	-	-	-

^1^ HGVS—Human Genome Variation Society; ^2^ CADD—Combined Annotation Dependent Depletion.

## Data Availability

The data presented in this study are sequencing data of human samples thus cannot be openly shared due to personal data protection.

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
