# Peer review of "Two Rare Variants in PLAU and BACE1 Genes—Do They Contribute to Semantic Dementia Clinical Phenotype?"

_genes, 2021, doi:10.3390/genes12111806_

Round 1
Reviewer 1 Report
The manuscript is very well written, and the findings are presented in a clear and adequate manner. The case is also well documented, as it provides an important direction for future research.
I only have a few minor comments/suggestions/questions:
- MAPT, PGRN and C9orf72 are well-established FTD risk loci, and readers are likely to be familiar with the topic; however, I believe it would be helpful if the authors stated this to reason the initial genetic approach.
- It is not clear to me why the authors selected puromycin, a translation inhibitor, rather than a more specific NMD inhibitor?
- Materials & Methods: please add the catalogue numbers of the primary antibodies used for Western blot analysis.
- Figure 1: please indicate the sites of atrophy, e.g. with arrows.
- Figure 2: “PLAU” mRNA in italics.
Author Response
Q1: MAPT, PGRN, and C9orf72 are well-established FTD risk loci, and readers are likely to be familiar with the topic; however, I believe it would be helpful if the authors stated this to reason the initial genetic approach.
A1: We thank the Reviewer for pointing this out. We have added the appropriate explanations in the text.
Q2: It is not clear to me why the authors selected puromycin, a translation inhibitor, rather than a more specific NMD inhibitor?
A2: The majority of small molecule NMD inhibitors exert their action through the inhibition of translation. We preferred puromycin over e.g. cycloheximide because puromycin can be diluted in water, while cycloheximide (CHX) is soluble in DMSO (or ethanol), which ultimately might affect the expression of many genes (see Verheijen et al. 2019; https://doi.org/10.1038/s41598-019-40660-0). For example, it is known that DMSO upregulates progranulin expression (Raitano et al. 2015, DOI: 10.1016/j.stemcr.2014.12.001).
As the Reviewer suggested, an alternative and more specific strategy could have been that of performing knock-out of SMG-1 and Upf1, key factors of NMD machinery. However, as we were using patients’ fibroblasts, the siRNA transfection efficiency of Upf1 was expected to be less efficient than using a small molecule, apart from being more time- and cost-consuming. As a result, our choice was to use puromycin which has been shown in recent studies to give very similar effects to CHX or even Upf1 silencing, see for example the recent manuscript by Qu at al. 2017 (DOI: 10.1016/j.mrfmmm.2017.09.005); Popp and Maquat 2015 (https://www.ncbi.nlm.nih.gov/pmc/articles/PMC4375787/). A small note has now been made in the revised version of the manuscript along with appropriate citations in the Materials & Methods section.
Q3:Materials & Methods: please add the catalogue numbers of the primary antibodies used for Western blot analysis.
A3: We have added the catalogue numbers of the primary antibodies used for Western blot analysis in Materials & Methods section.
Q4:Figure 1: please indicate the sites of atrophy, e.g. with arrows.
A4: We have added arrows in Fig 1B indicating the sites of atrophy, according to the Reviewer’s suggestion.
Q5: Figure 2: “PLAU” mRNA in italics.
According to the reviewer’s suggestion we have made the necessary corrections.
Reviewer 2 Report
I would like to start by congratulating the authors on the very interesting and well characterized case report. A deeper knowledge of genetic dementias is of great importance.
The authors provide an study of the identified variant om PLAU and BACE2 genes.
I would just have a few comments:
- Was the age of first symptoms 64yo ?
- Cortical functions examination, as assessed by neuropsychology tests, is part of the neurological examination. The authors should not state the patient had a normal neurological examination at age 69 if he was demented and also aphasic
- Symptoms progression is a bit confusing, and mixed with the results from MRI and PET. It would be clearer if the authors describe all clinical progression first and then detail the performed exams.
- There was no information on progression from 73 till 76
- Family history should be described in the case presentation. Also family tree should be part of the main documents and not as supplementary material
- Was the patient’s living brother examined? He also had memory complaints
- CSF biomarkers would be of great importance…
- Lastly, there should be great caution when describing potentially new genes. The authors have been careful in their discussion, but it is never too much to emphasize this
Author Response
Q1: Was the age of first symptoms 64 years?
A1: First symptoms appeared when the patient was 64 years old and we indicated it in the text, according to the Reviewer’s suggestion.
Q2:Cortical functions examination, as assessed by neuropsychology tests, is part of the neurological examination. The authors should not state the patient had a normal neurological examination at age 69 if he was demented and also aphasic
A2: At our center, a neuropsychologist assesses cognitive functions and also conducts a detailed cognitive and behavioral interview. Neuropsychological assessment was planned, executed, and interpreted by a neuropsychologist and was not a part of neurological examination. That's why these results are provided independently. According to the Reviewer's suggestions, we have rephrased the sentence on neurological examination in the case description. Semantic assessment and the differential diagnosis of fluent aphasia is not a part of the general neurological examination and semantic deficits were not apparent in the neurological interview.
Q3:Symptoms progression is a bit confusing, and mixed with the results from MRI and PET. It would be clearer if the authors describe all clinical progression first and then detail the performed exams.
A3: We have reorganized the case description according to the Reviewer’s suggestions.
Q4:There was no information on progression from 73 till 76
A4: The patient was institutionalized in a long-term care unit at the age of 73 and remained there till his death. The patient was treated with neuroleptics at the beginning of the stay at a long-term care unit. We did not think a psychological or neurological examination would be particularly helpful in such a situation. Moreover, we could not see the patient during the pandemic due to COVID-19 restrictions. The patient died during the COVID-19 pandemic.
Q5:Family history should be described in the case presentation. Also, the family tree should be part of the main documents and not as supplementary material
A5: More information concerning the patient’s family history was included in the case description and the family tree, in line with the Reviewer’s suggestion. The family tree has been moved to the main manuscript and became Fig1 A.
Q6:Was the patient’s living brother examined? He also had memory complaints
A6: We included the description of the neuropsychological examination of the patient’s brother (the one with mild cognitive impairment with the predominant semantic deficit) in the supplementary file as Table S5. The examination was conducted once. Unfortunately, the brother lived far away from our center and agreed to come only once.
Q7:CSF biomarkers would be of great importance…
A7: Yes, we are aware of that. Unfortunately, CSF biomarkers are not routinely assessed in Poland. At our center, this would require hospitalization. This test was not available in our center during the patient's first visits. Later on, due to the severity of behavioral problems, we did not think that hospitalization at our inpatient unit was feasible.
We added this point as a study limitation in the discussion.
Q8:Lastly, there should be great caution when describing potentially new genes. The authors have been careful in their discussion, but it is never too much to emphasize this.
A8: Both PLAU and BACE1 have been previously associated with Alzheimer’s disease, which has been described in the discussion. In the discussion, we stressed that the contribution of PLAU and BACE1 variants to early-onset dementia cases should be assessed taking into account co-segregation with the disease phenotype.